# Dating Skin Lesions of Forensic Interest by Immunohistochemistry and Immunofluorescence Techniques: A Scoping Literature Review

**DOI:** 10.3390/diagnostics14020168

**Published:** 2024-01-11

**Authors:** Luca Tomassini, Massimo Lancia, Roberto Scendoni, Anna Maria Manta, Daniela Fruttini, Erika Terribile, Cristiana Gambelunghe

**Affiliations:** 1International School of Advanced Studies, University of Camerino, 62032 Camerino, Italy; 2Forensic Medicine, Forensic Science and Sports Medicine Section, Department of Medicine and Surgery, University of Perugia, Piazza Lucio Severi, 06132 Perugia, Italy; massimo.lancia@unipg.it (M.L.); erika.terribile@studenti.unipg.it (E.T.); cristiana.gambelunghe@unipg.it (C.G.); 3Department of Law, Institute of Legal Medicine, University of Macerata, 62100 Macerata, Italy; r.scendoni@unimc.it; 4Department of Anatomical, Histological, Forensic Medicine and Orthopedic Sciences, Sapienza University, 00185 Rome, Italy; annamaria.manta@uniroma1.it; 5Section of Internal Medicine and Endocrine and Metabolic Sciences, Department of Medicine and Surgery, University of Perugia, 06123 Perugia, Italy; daniela.fruttini@unipg.it

**Keywords:** wound age estimation, biochemical methods, immunohistochemistry, immunofluorescence, forensic diagnosis

## Abstract

Wound age estimation is a significant issue in forensic pathology. Although various methods have been evaluated, no gold standard system or model has been proposed, and accurate injury time estimation is still challenging. The distinction between vital skin wounds—i.e., ante-mortem lesions—and skin alterations that occur after death is a crucial goal in forensic pathology. Once the vitality of the wound has been confirmed, the assessment of the post-trauma interval (PTI) is also fundamental in establishing the causal relationship between the traumatic event and death. The most frequently used techniques in research studies are biochemistry, molecular biology, and immunohistochemistry (IHC). Biochemical methods take advantage of the chemical and physical techniques. A systematic literature search of studies started on 18 February 2023. The search was conducted in the main databases for biomedical literature, i.e., PubMed and Scopus, for papers published between 1973 and 2022, focusing on different techniques of immunohistochemistry and immunofluorescence (IF) for estimating the PTI of skin wounds. The present study involves a comprehensive and structured analysis of the existing literature to provide a detailed and comprehensive overview of the different IHC techniques used to date skin lesions, synthesize the available evidence, critically evaluate the methodologies, and eventually draw meaningful conclusions about the reliability and effectiveness of the different markers that have been discovered and used in wound age estimation.

## 1. Introduction

Wound age estimation is a significant issue in forensic pathology. Although various methods have been evaluated, no gold standard system or model has been proposed, and accurate injury time estimation is still challenging [1]. The distinction between vital skin wounds—i.e., ante-mortem lesions—and skin alterations that occur after death is a crucial goal in forensic pathology. Once the vitality of the wound has been confirmed, the assessment of the post-trauma interval (PTI) is also fundamental in establishing the causal relationship between the traumatic event and death [2].

With the development of new laboratory techniques, it is now possible to integrate the study of the age of the lesions with multiple approaches, including morphological, cytohistological, biochemical, and genetic analysis. The most frequently used techniques in research studies are biochemistry, molecular biology, and immunohistochemistry (IHC) [3,4,5,6,7,8,9,10,11,12].

Among all of the techniques, immunohistochemistry provides a great deal of evidence in the literature and, therefore, it is a valuable choice in determining if a lesion is vital or not, thanks to a wide variety of markers (tissue molecules, cytokines, and growth factors) [13,14]. Furthermore, IHC has proven to be more useful, not only because of its ease of use and high reliability, but, above all, because of its ability to analyze the localization of the molecules of interest, especially when compared to other techniques [15].

The existing literature on IHC techniques for wound age estimation has produced diverse and sometimes conflicting results.

The present study involves a comprehensive and critical review of the existing literature to provide a detailed and structured overview of the different IHC techniques used to date skin lesions, to synthesize the available evidence, to critically evaluate the methodologies, and, finally, to draw meaningful conclusions about the reliability and efficacy of the different markers that have been discovered and used in wound age estimation. This analysis can contribute to the advancement of knowledge in forensic science, potentially leading to improved forensic investigation practices, as well as a more accurate determination of the timing of skin lesions.

## 2. Materials and Methods

A systematic literature search of studies started on 18 February 2023. The search was conducted in the main databases for biomedical literature, i.e., PubMed and Scopus, for papers published between 1973 and 2022, focusing on different techniques of immunohistochemistry and immunofluorescence (IF) for estimating the post-trauma interval (PTI) of skin wounds. 

The aim of this study was to evaluate different targets and existing techniques for the dating of skin lesions of different natures for forensic use.

The generic free-text search terms were: “(((timing of wounds) OR (Skin wounds age estimation) OR (bruising timing) OR (bruising age estimation) OR (dating of bruising)) AND ((forensic pathology) OR (legal medicine))) AND ((histopathology) OR (immunohistochemistry) OR (histochemistry) OR (technique)).”

The following parameters were used as inclusion criteria:Language: full-text written in EnglishType of article: original articles, prospective cohort studies, retrospective cohort studiesObjective: determine PTI using different targets by both immunohistochemical and immunofluorescence methods (clearly stated in the title, abstract, methods, and/or keywords)Biological sample: skin tissue from cadavers, post-mortem studies, biopsies on living humans, but also prospective studies on experimental animals such as mice, rats, and porcine


The exclusion criteria were the following:Language: non-EnglishType of article: reviews and systematic reviewsTechnique: wound-age estimation techniques other than immunohistochemistry

After the first screening, the following data were recorded from the chosen papers:-Authors, the country where the study was performed, date of publication, markers, study design, and aim.-Inclusion and exclusion criteria, control size, sample size, sample characteristics, the origin of the sample, type of injuries, material and methods, and the type of article.-Statistical analysis: in 15 articles there was no statistical study, so only the authors’ observations and semi-quantitative results were included.

The current scoping review followed the Preferred Reporting Items for Systematic Review and Meta-Analysis (PRISMA) guidelines.

Reporting Items for Systematic Review and Meta-Analysis (PRISMA). We thoroughly read the bibliographies of the identified papers and cross-referenced them to identify additional relevant literature. Using PRISMA standards, each study underwent a methodological assessment, including an assessment of bias.

The data collection process included both study selection and data extraction.

Articles with titles or abstracts that met the inclusion criteria were independently assessed by three researchers who resolved disagreements by consensus. Two researchers performed the data extraction, which was then reviewed by two other researchers, and then reconfirmed by an additional pair of investigators. Institutional review board approval was deemed unnecessary for this study, as it did not involve human subjects. 

The results of the statistical approach came from statistical tests (linear coefficient of determination (R), standard deviation, mean squared error, prediction/correlation formulas, 95% confidence interval (95%CI), Scheffè’s test, Dunnett’s test, chi-squared test, ANOVA, Mann-Whitney test, Bonferroni post-hoc, Kruskal-Wallis test, Fisher’s test). 

In addition, during the review of the studies of interest, 17 other papers relevant to the present research were extracted from their bibliographies. A summary table (Appendix A (studies on human) and Appendix A (studies on animal model)) of the results was prepared based on the information collected on each of the articles of interest.

## 3. Results

Of the 326 articles initially identified and processed, consisting of 288 from PubMed and 38 from Scopus, 29 studies were included, plus those extracted from their bibliographies [16], for a total of 46 articles. In detail, from the original 326 articles, with the inclusion and exclusion criteria mentioned above having been applied, 19 were non-English studies, 14 were reviews, 235 articles investigated techniques other than IHC and IF or target tissues other than skin, and 29 were duplicates (Figure 1). Regarding the country of the included studies, 25 (54.3%) were conducted in Europe, mainly in Germany (39.1%), Italy (6.5%), the Netherlands (4.3%), France (2.2%), and Denmark (2.2%), and the remainder were conducted in Japan (28.3%), China (8.7%), Egypt (6.5%), and Iraq (2.2%). In total, 32 retrospective cohort studies, nine prospective cohort studies, and five both prospective and retrospective studies were analyzed. In the case of experimental animals, all procedures were performed according to the protocol approved by the Institutional Animal Care and Use Committee; in the case of living patients, they gave written consent after being informed of the purpose of the study. In the cohort studies, 34 human skin wounds were obtained from autopsies, one from a live biopsy, and eight from surgical or operative procedures. There was also a wide variety of markers in these cohort studies, with 25 being multi-markers and 21 being monomarkers. All markers, alone or in combination, were cytokines, specific immune proteins and cells, different types of collagens, actin, selectins, histamine, fibronectin, cytokeratin, ubiquitin, caspases, coagulation factors, growth factors, aquaporins, cyclooxygenase, lipase, vitronectin, tenascin, peptidase and phosphatase, metalloproteinase, and some types of progenitor cells. (Appendix A provides details of the studies analyzed regarding humans, while Appendix A summarizes the studies conducted on animal models).

In terms of technique, 32 (70%) studies used IHC alone, five (10%) used IF alone, and the remaining nine (20%) used a combination of the two. In addition, 31 of the 46 studies included a statistical analysis with statistically significant results, whereas 15 studies proposed qualitative observations.

## 4. Discussion

Wound age estimation remains a challenge for forensic pathologists worldwide, as post-mortem examination of dead bodies can reveal a multitude of injuries, the time of production of which is only imaginable [17]. In one of the most recent reviews of the literature, Li et al. examined publications from 2010–2016 and concluded that there are no standardized and reliable biomarkers for estimating wound age, although many molecules have been studied [16]. Therefore, the use of a panel of markers may be more reliable in dating wounds than the study of a single marker. 

Once a wound is created, the process of wound healing begins immediately, with rapid haemostasis, appropriate inflammation, mesenchymal cell differentiation, proliferation and migration to the wound site, appropriate angiogenesis, rapid re-epithelialization (re-growth of epithelial tissue over the wound surface), and proper synthesis, cross-linking, and orientation of collagen to provide strength to the healing tissue.

Temporal trends of the main markers identified in the literature are summarized in Figure 2.

### 4.1. Coagulation Factors

The first step in the healing cascade is rapid haemostasis. As reported, Van De Goot et al. developed the Wound Age Probability Scoring System based on immunohistochemical expression levels of fibronectin, CD62p, and factor VIII in wound bleeding. In recent wounds, there was a significant increase (*p* < 0.001) in the IHC score for all three markers compared to unwounded control skin samples [18].

Gauchotte et al. investigated FVIIIra, CD15, and tryptase to interpret the timing of skin puncture wounds. FVIIIra showed high sensitivity (100%) but lower specificity (47%), with overexpression observed in both surgical and negative control wounds (53%). Consequently, FVIIIra is a highly sensitive but non-specific marker of skin wound vitality, whereas CD15 and tryptase are considered more useful for distinguishing recent ante-mortem from post-mortem injuries [19].

As highlighted in Figure 2, these studies showed statistically significant results with both fibronectin/CD62p/factor VIII and FVIIIra/CD15/tryptase complexes tending to stain positive in early skin injury, particularly at a PTI of 0–1 h.

### 4.2. Proinflammatory Cytokines

Inflammatory cytokines, including IL-1, IL-6, IL-8, and TNFα, play crucial roles in the inflammatory phase of wound healing. A cohort study by Kondo et al. (1996) highlighted statistically significant peaks of proinflammatory cytokines such as TNFα and IL-1β in 3 h PTI, with a smaller but still relevant IL-1β peak after 6 h, and an IL-6 peak after 12 h post-trauma [20]. Neutrophil infiltration is positive for IL-10, IL-11, IL-6, and TNFα between 3 and 6 h, while phagocytes are positive for IL-10, IL-11, IL-6, and TNFα during the transitional phase (24 h) [20].

In a retrospective cohort study by Kondo et al. (1999), the temporal expression of interleukin-1α (IL-1α) in human skin wounds was investigated, confirming earlier findings regarding the presence of polymorphonuclear cells as early as 4 h post-trauma. The cytoplasm of these neutrophils showed positive staining with anti-IL-1α antibody, which persisted in wounds up to one day old. The study also suggested that proportions of IL-1α-positive cells greater than 30% could indicate a wound age of one day or less [21].

Grellner et al. [22] investigated the correlation between IL-1, IL-6, TNF-α, and age of sharp force injuries, and showed the early appearance of proinflammatory cytokines in vital skin wounds with different expression levels and distribution patterns compared to intact skin. The earliest response occurred 15 min after injury and persisted for several hours [22].

In the study by Kondo et al. (2002), the investigation of IL-8, monocyte chemoattractant protein (MCP)-1 and macrophage inflammatory protein (MIP)-1α in human skin wounds showed that neutrophils with IL-8, MCP-1, or MIP-1α were initially predominant at the wound site for up to 12 h after injury. As wound healing progressed, macrophages took over and IL-8, MCP-1, and MIP-1α were then localized in the cytoplasm of macrophages and fibroblasts [23].

### 4.3. Immunity Cells and Factors

The prospective study by Barington et al. in the porcine model showed that the severity of the bleeding, as well as the amount of necrotic muscle tissue, were highly dependent on the impact force and the presence of neutrophils and macrophages in the dermis at 2 h, regardless of the force used [24]. Otherwise, the average number of neutrophils in the subcutaneous tissue showed a time- and force-dependent response [24].

Based on immune cell studies, particularly neutrophils, macrophages, and lymphocytes, the research by Fronczek et al. demonstrates a time-dependent pattern in wound healing. Neutrophil count and MIP-1-positive inflammatory cells are significantly higher in 0.2–4 days old wounds, as observed in their skin biopsy study on living subjects. Additionally, IL-8 expression in the epidermis doesn’t correlate with wound age, while activated macrophages (CD68-positive) peak in 2–4 day-old wounds, and CD45-positive lymphocytes are highest in 0.2–2 day-old wounds, declining in wounds up to 10 days old [25].

Yagi et al. conducted a prospective and retrospective study evaluating the expression of CD32B, CD68, and CD14 to assess wound age in murine and human skin wounds. CD14 showed a significant increase between 2–5 days after injury, with 100% sensitivity and 87.2% specificity in detecting wounds at 1–5 days. The combination of CD14/CD32B/CD68 expression indicated a wound age of 1–5 days with high specificity, while CD14-/CD32B-/CD68- indicated a wound age of less than one day [26].

Fouad et al. found that a post-traumatic interval (PTI) of 1-3 days showed peak CD14 expression (96.40 ± 3.78%), indicating extensive infiltration of dermal macrophages and lymphocytes. Conversely, wounds older than 3 days showed a significant decrease in CD14 expression (14.80 ± 3.49%) with a limited presence of inflammatory cells and fibroblasts. The study concludes that immunohistochemistry (IHC) with CD14 is a reliable marker for determining wound age, particularly in wounds aged 1–3 days, with an overall accuracy of 100% [27].

In a recent study by Kuninaka et al. using anti-CD11c and -HLA-DRα antibodies to study dendritic cells in human skin wounds, it was found that all wound samples, regardless of heterogeneity, had a substantial number of CD11c+/HLA-DRα+ dendritic cells, with a mean of 57.7 ± 3.6 cells [28]. The results suggested that a count of >50 DCs in wound samples would indicate a wound age of between 4 and 14 days. This time range can be narrowed by using combined evaluation with other markers [28].

Mast cells, which have a longer lifespan than normal cells, have been studied in addition to histamine markers. In a study by Jebur et al., mast cell tryptase, IL-1 beta, and IL-6 were used to assess wound vitality and age in human skin wounds, and showed a significantly higher mast cell density in the dermis of the sample group compared to the control group [29].

Immunohistochemical analysis revealed a direct correlation between mast cell infiltration and time since wounding in the study group.

Bonelli et al. focused on mast cells in vital wounds and observed a peak in mast cell numbers after 1–3 h, but also an alteration in post-mortem cases, which showed significantly fewer mast cells compared to other groups [30].

In these studies, Zhong and Zhen in 1991 studied histamine both in human and rat skin lesions and found that histamine content gradually increases up to 30 min after injury [31].

VCAM-1, a member of the immunoglobulin superfamily, is a specific marker because of its variable expression on endothelial cells. VCAM-1 requires activation by lipopolysaccharides and cytokines, such as IL-1β and TNF-α, released during wound healing. Its role is to regulate the diapedesis of lymphocytes, monocytes, and eosinophils from blood vessels into tissues [32].

Dressler et al. (1999) discovered time-dependent VCAM-1 expression in skin wounds, with strong positive staining observed at 3 h and 3.5 days after injury. The intensity of expression peaked at 4–6 h and then declined to baseline levels. Staining patterns differed between post-mortem and vital wounds, suggesting VCAM-1 as a potential marker for assessing wound vitality. However, reliance on VCAM-1 expression alone is inadequate for accurate estimation of wound age [33].

This category is very useful for dating in the range of 1 h to 5 days, except for dendritic cell count > 50, which occurs up to 14 days.

### 4.4. Growth Factors

Grellner et al. investigated TGF-α and TGF-β1 in human skin wounds, observing their initial responses within 10 min, and a marked response within 30–60 min, using immunohistochemical methods. Although TGF-α and TGF-β1 do not correlate, TGF-α, with homology to epidermal growth factor (EGF), plays a crucial role in stimulating and regulating angiogenesis in the early post-injury period, contributing to the assessment of wound age and vitality [34].

Moreover, in the more advanced stages of healing vascular endothelial growth factor (VEGF) is produced by many cell types involved in wound repair: endothelial cells, fibroblasts, smooth muscle cells, platelets, neutrophils, and macrophages [35].

Hayashi et al. investigated the time-dependent expression of VEGF, and neutrophils without immunopositivity for VEGF were observed in wound samples aged from 4 h to 1 day. Therefore, a VEGF-positive ratio of more than 50% may indicate a wound age of 7 days or more, up to 14 days [36]. The results are consistent with the study by Khalaf et al. in rats, suggesting CD68, α-SMA, VEGF, and TGF-β1 as potential biomarkers for determining wound age, with peak expression observed at 5 days and a maximum at 7 days [37].

### 4.5. Enzymes

Proteolytic enzymes are a family of proteins that are used to break down necrotic cellular debris. They are often produced endogenously as precursor proteins whose activation is precisely regulated [38].

In the work of Betz et al., enzyme histochemistry was evaluated to assess its applicability in wound age estimation [39]. Non-specific esterase was found in 40% of samples of vital human skin wounds with a PTI between 1 h and 5 days. ATPase, aminopeptidase, and alkaline phosphatase were found in approximately 20% of samples with a PTI between 4 h and 5 days [39].

In skin wounds, matrix metalloproteinases are synthesized by keratinocytes, fibroblasts, endothelial cells, and macrophages, and are upregulated within hours of injury. Collagenases, gelatinases, and stromelysins are particularly important in wound healing: MMP-1, MMP-3, MMP-9, and MMP-10 contribute to keratinocyte migration during epithelialization [40,41].

Ishida et al. investigated MMP-2 and MMP-9 in human skin wounds, and found that MMP-2+ macrophages increased significantly with wound age, suggesting a wound age of 9–12 days when MMP-2+ cells exceeded 25, while MMP-9+ macrophages increased significantly in wounds aged 4–14 days, suggesting a wound age of 3–14 days when the number of MMP-9+ cells exceeded 30 [42].

Monoacylglycerol lipase, a key enzyme in the hydrolysis of the endocannabinoid 2-arachidonoylglycerol (2-AG), was found to show a time-dependent correlation with the expression of wound injury in mice. The enzyme converts monoacylglycerols into free fatty acids and glycerol [43].

Ishida et al. examined COX-2 expression in wounds of different ages and found that in unwounded skin, as well as wounds less than 30 min old, there was minimal leukocyte recruitment with only a few resident cells expressing COX-2. Wounds aged 2 h to 2 days showed a prominent presence of myeloperoxidase-labelled neutrophils expressing COX-2. The study suggests that a COX-2 positive ratio > 40% is indicative of a wound age between 8 h and 3 days, and ratios above 50% may have forensic significance [44].

### 4.6. Glycoproteins and Basal Membrane Components

Up to 14 different collagen subtypes have been identified, and at least 6 are identifiable in skin. They can be subdivided into interstitial collagens (type I, III, V, and VI) and specific basement membrane collagens (type IV and VII) [45].

Betz et al. investigated the location of interstitial collagen type V and compared these findings with the presence and location of collagen III [46].

Immunohistochemically, collagen III was undetectable in wounds less than 2.5 days old and gradually increased in intensity in wounds older than 5 days. Collagen V appeared three days after wounding and was less intense than collagen III. Both types of collagen were present in the oldest wounds [46].

Regarding interstitial collagen, Collagen types I and VI consistently appear around 6–7 days post-injury in human skin wounds, and their absence may indicate a wound age of less than 6–7 days, according to Betz et al.’s evaluation for forensic purposes [47].

Collagen VII, found exclusively in the basement membrane of the epidermis and other stratified epithelia, was studied in heterogeneous human skin wounds by Betz et al. Fragments of the basement membrane appeared around 4 days after injury, while complete rearrangement took 8 days or more [48].

Betz et al. demonstrated that myofibroblasts cannot precisely determine the age of older wounds, but they discovered the appearance of Alpha-SMA-positive myofibroblasts at around 5 days post-injury. Therefore, the presence of myofibroblasts can estimate a wound age of approximately 5 days or more [49].

During wound healing, fibroblasts can transform into so-called myofibroblasts, a subtype of fibroblast cells characterized by the presence of various cytoplasmic filaments such as vimentin, alpha-smooth muscle actin, and desmin. In addition to the transient expression of alpha-smooth muscle actin, myofibroblasts can synthesize several extracellular matrix components during wound healing, such as collagen type IV, laminin, and HSPG [48,50].

The findings of Betz et al. showed that the pericellular appearance of laminin and HSPG around myofibroblasts indicates a wound age of at least 1.5 days [51]. Laminin seems to be expressed in a larger number of cells and wounds as compared to HSPG rendering the immunohistochemical localization of laminin a more useful parameter in this context [51].

Fibronectin is also a potent chemoattractant mainly involved in tissue repair processes such as wound healing and for these reasons was investigated also for forensic purposes [52,53].

Legaz et al. demonstrated that vital wounds showed strong fibronectin-positive reactions in basement membranes and interstitial connective tissue within a post-mortem interval of 19–36 h. Fibronectin was weakly positive in the basement membrane of the dermo-epidermal junction and in the vasculature of healthy skin. Previous literature suggests fibronectin as a potential vitality marker for wounds with a survival time of more than a few minutes [52,53,54].

Grellner et al. conducted immunohistochemical investigations on porcine skin wounds, attributing fibronectin-positive immunoreaction in post-mortem alterations to passive transudation of blood components from injured vessels. Tenascin, interacting with fibronectin, supports cell adhesion and exhibits immunomodulatory activities. The intensity of tenascin staining decreases with wound age and remains visible in wounds around 1.5 months old [55,56].

The selectins, including E-selectin found in endothelial cells, are a family of cell adhesion molecules (CAMs) involved in leukocyte rolling and adhesion to activated endothelial cells. E-selectin staining was observed in wounds from 1 h to 17 days post-injury, with a significant decrease at 12 h post-injury [57,58,59].

### 4.7. Other Markers

Crucial events in wound healing, including re-epithelialization, neovascularization, and extracellular matrix production, involve processes such as cell migration and proliferation [60,61].

Recent studies propose new roles for aquaporins (AQPs), known as water transport channel proteins, in wound healing, with evidence indicating their involvement in the migration of various cell types such as keratinocytes, endothelial cells, astroglial cells, and kidney proximal tubule cells [62,63].

Ishida et al. investigated the relationship between the percentage of AQP1 and AQP3 and wound age and found that AQP3+ keratinocytes were present in 4–14 day old wounds, whereas an AQP3+ cell count of >300 indicated a wound age of 5–10 days [64].

Nitric oxide (NO), produced by nitric oxide synthase (NOS) and inducible nitric oxide synthase (iNOS), serves as a critical marker of M1 macrophage activation. While iNOS is known for its antimicrobial effects, its role in the regulation of immune responses and M1 macrophage differentiation is not fully understood, despite a growing body of evidence [65,66].

Oxygen-regulated protein 150 (ORP150), an inducible endoplasmic reticulum (ER) chaperone, is upregulated in response to cellular stress and plays a cytoprotective role in ischemia-reperfusion injury models; the highest average ORP150-positive ratio was noted in wounds aged 7–14 days, suggesting that a positive ratio >50% is a strong indicator of a wound age within this timeframe [67].

Ubiquitin (Ub), a highly conserved protein, facilitates non-lysosomal protein degradation by covalently attaching to various proteins through the Ub protein ligase system. The study by Kondo et al. in mice and human cadavers revealed time-dependent ubiquitin expression, highlighting early wound responses with neutrophil infiltration and strong intranuclear Ub-positive reactions at wound sites [68,69].

Chitinase-3-like protein 1 (CHI3L1 or YKL-40) is a secreted glycoprotein of approximately 40 kDa encoded by the CHI3L1 gene. Studies, such as those by Murase et al. in murine skin wounds showing temporal changes in the expression of chitinase-3-like protein 3 (CHI3L3), provide a basis for investigating the variable expression of other proteins in the chitinase and chitinase-like protein (C/CLP) family, to which CHI3L3 belongs, during wound healing [70,71].

Murase et al. investigated CHI3L1 immunohistochemical expression in both human and murine skin wounds. In murine wounds, CHI3L1 expression changed over time. In human cadaver skin wounds, weak CHI3L1 expression (0.11 ± 0.024) was observed on days 0 to 1. From day 4 to 6, there was a significant increase in CHI3L1-expressing cells (5.35 ± 0.35), followed by a decrease in expression from day 7 to 13 (1.53 ± 0.24) [72].

Ishida et al. investigated the potential of endothelial progenitor cells (EPCs) for wound age determination. EPCs were first observed in human skin wounds two days after injury, and their number progressively increased as the wound aged [73,74].

In murine wounds, CD34+/Flk-1+ endothelial progenitor cells (EPCs) were barely detectable in early post-mortem changes (0–1 day). Wounds aged 2–6 days had few EPCs (<20), whereas wounds aged 7–12 days showed a significant increase in EPCs (>20), suggesting a possible correlation between the number of EPCs and wound age [74].

These markers are particularly useful for dating skin wounds up to 1-day PTI, also depending on the percentage present for greater dating accuracy.

Overall, this category covers a range from 2 to 14 days.

## 5. Conclusions

Wound age estimation is a critical issue in forensic pathology. Pathologists need sensitive and specific markers to make an objective assessment and to provide scientifically validated evidence.

The present study highlights that the studies using multiple markers and techniques show higher sensitivity and specificity but are more time-consuming and complex.

While many observational studies highlight the need for additional case study research, some articles suggest correlations between post-mortem wound vitality and specific markers that require further confirmation through larger case series. In conclusion, immunohistochemical and immunofluorescence techniques are effective in monitoring lesion progression and vitality but cannot accurately determine the post-mortem interval.

In conclusion, the comprehensive analysis of various markers related to coagulation factors, proinflammatory cytokines, enzymes, glycoproteins, basement membrane components, interstitial collagen, myofibroblasts, and other markers provides valuable insights into the forensic estimation of skin wound age. The complex interplay of these markers provides a nuanced understanding of the temporal progression of skin injury.

The fibronectin/CD62p/factor VIII and FVIIIra/CD15/tryptase complex staining patterns show statistically significant results, particularly within the first hour of injury. Proinflammatory cytokines provide a useful timeframe for dating from 1 h to 5 days, except for dendritic cell count > 50, which extends the dating range up to 14 days.

Enzymes such as COX-2 are indicative markers, with a positive ratio of >40% suggesting a wound age between 8 h and 3 days, and ratios above 50% potentially having forensic significance. Glycoproteins such as laminin and HSPG, as well as basement membrane components, exhibit distinct patterns that contribute to the estimation of wound age. The appearance of collagen types III and V, together with the presence of collagen types I, VI and VII, further refines the dating process and allows differentiation between wounds of different ages.

The role of myofibroblasts, particularly alpha-SMA-positive myofibroblasts, emerges as a key indicator, with their appearance at around 5 days post-injury helping to estimate wound age. Markers such as AQP, ORP150, and ubiquitin contribute to the overall dating accuracy over a wider range of 2 to 14 days, while also being detectable in earlier wounds.

In summary, the integration of these diverse markers enhances the forensic toolkit for assessing the age of skin wounds, providing forensic investigators with a more detailed and accurate understanding of the temporal progression of injuries. The multifaceted approach presented in these studies provides a valuable foundation for advancing forensic practice in the area of wound age estimation.

Overall, the use of a multi-marker approach can be considered the most reliable method for accurately estimating the age of a wound. However, future research is needed to identify a highly sensitive and specific marker for post-mortem wound age estimation.

## Figures and Tables

**Figure 1 diagnostics-14-00168-f001:**
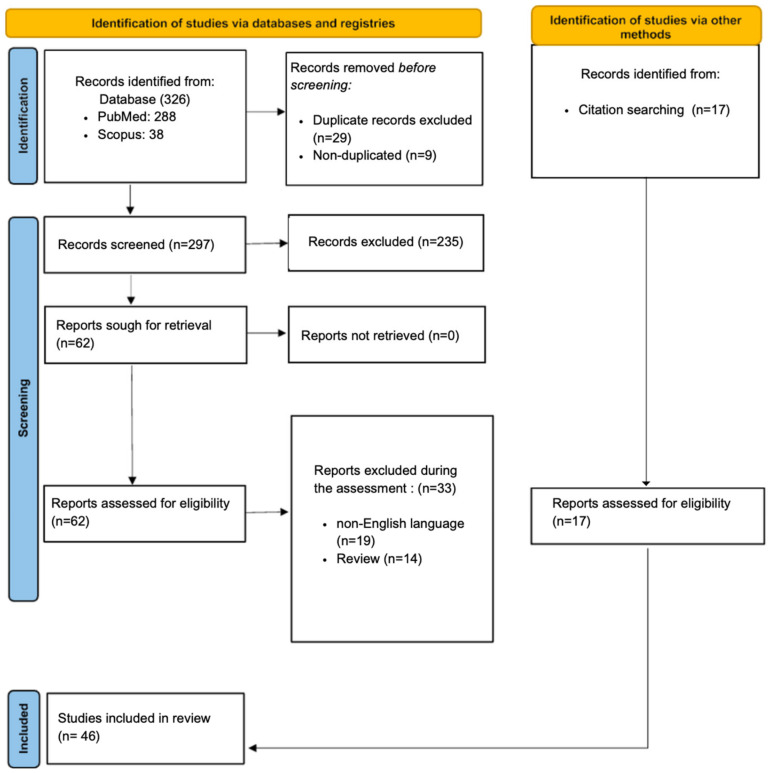
Descriptive diagram of the paper selection process.

**Figure 2 diagnostics-14-00168-f002:**
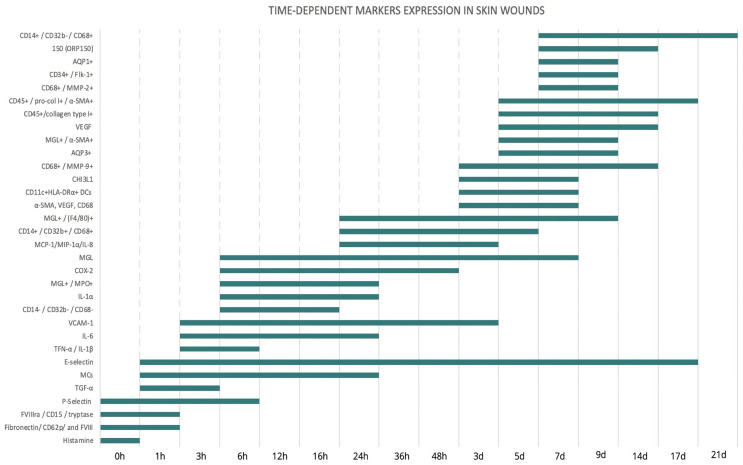
Temporal trends of the main markers for dating lesions.

## Data Availability

Not applicable.

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
