# Peer review of "Dating Skin Lesions of Forensic Interest by Immunohistochemistry and Immunofluorescence Techniques: A Scoping Literature Review"

_diagnostics, 2024, doi:10.3390/diagnostics14020168_

Round 1

Reviewer 1 Report

Comments and Suggestions for Authors

Review of “

Dating skin lesions of forensic interest by immunohistochemistry and immunofluorescence techniques: A systematic literature review.” by Tomassini et al. 

Summary:

This systematic review aims to analyze the existing literature to provide a detailed and comprehensive overview of the different IHC techniques used to date skin lesions in order to draw meaningful conclusions about the reliability and effectiveness of different markers used in wound age estimation.

 Reviewer’s major comment:

This work doesn’t fulfil the criteria of a systematic review and is rather in the form of a scoping review or an evidence map of the subject. Mainly, the reporting of bias of the included studies is missing, and it is unclear how many reviewers collected data from each article and if they worked independently. The inclusion of case reports in this systematic review is questionable considering their associated low evidentiary value. In addition, it is unclear if the generic free text search terms will locate all studies in the field, information on selected MeSH terms is missing and limited databases are searched. I would suggest to remove systematic review from the title and in the manuscript.

Anyway, this work is useful as an evidence map or summary of dating skin lesions with IHC techniques and has potential to provide important information in the field.

 Reviewer’s comments:

 1) The manuscript would benefit from a better structure where the long introduction is shortened and focused to be more concise.

·        2) Figure 1. should be at the beginning of the results, before table 1.

·        3)  Please, divide the long table 1 into 2 separate tables: one for animal studies and the other with human studies and discuss the results accordingly. Consider adding these as supplementary materials.

·         4)It is mentioned in the result section that 31 out of 46 studies show significant results, these studies should be highlighted in the tables since this seems to be the results worth making conclusions from.

·        5) Figure 2 needs to be in a higher resolution, it is difficult to see the text on the Y-axis. This figure seems to be of importance and should contain the proven results (significant findings).

·       6)  Consider structuring the discussion to reflect figure 2. Suggest that markers are grouped according to their putative timeframes (e.g 0-6 hours; 06-24 hours etc..) instead of grouped according marker type.

·        7) The reader is left without any clear conclusions regarding which markers or combination of markers that are useful when dating skin lesions, and if there are any recommendations regarding how the sampling should be performed. The conclusion could be formulated around the findings and the use of the findings, a summary figure similar to figure 2 would be constructive to show the recommended markers/combination of markers (only significant data included).

Comments on the Quality of English Language

The manuscript would benefit from from language editing 

Author Response

Dear Mr. Dusan Vukelic,

We would like to thank you for allowing us to submit a revised draft of our manuscript titled Wound age estimation by immunohistochemistry and immunofluorescence techniques: A systematic literature review.

We deeply appreciate the time and effort that you and the reviewers have dedicated to providing your valuable feedback on our manuscript. We are grateful to the reviewers for their insightful comments on our paper.

Following the reviewer’s suggestions, we have made some changes to our manuscript.

Here is a point-by-point response to the reviewers’ comments and concerns.

Reviewer #1 Response

This work doesn’t fulfil the criteria of a systematic review and is rather in the form of a scoping review or an evidence map of the subject. Mainly, the reporting of bias of the included studies is missing, and it is unclear how many reviewers collected data from each article and if they worked independently. The inclusion of case reports in this systematic review is questionable considering their associated low evidentiary value. In addition, it is unclear if the generic free text search terms will locate all studies in the field, information on selected MeSH terms is missing and limited databases are searched. I would suggest to remove systematic review from the title and in the manuscript.

We apologize if the systematic approach was not clearly outlined. Thanks to your comment, we realized that we forgot to include more information regarding our approach in this review. In particular, the evaluation of bias was left implied, while we are aware that it is crucial in this type of research. Moreover, regarding the inclusion of case reports, at first, we intended to add them but they were ultimately excluded from the analysis (as you can see from the tables we first provided, there were no case reports). The search terms locate all the studies in the field of our interest. The two databases were chosen as they are the most commonly used in our practice so we think that they provide more than enough data on the matter.

Anyway, this work is useful as an evidence map or summary of dating skin lesions with IHC techniques and has potential to provide important information in the field.

We thank you for valuing our research and recognizing the implications of such a study.

  • The manuscript would benefit from a better structure where the long introduction is shortened and focused to be more concise.

We agree with your observation. We had the intent to portray all the techniques but, as you have kindly highlighted, it may digress from our primary scope. We have now eliminated the unnecessary information, as you can see in the Introduction section.

  • Figure 1. should be at the beginning of the results, before table 1.

We too noticed that the final layout of our manuscript was not optimal. Therefore, the location of the figures and tables has been changed.

  • Please, divide the long table 1 into 2 separate tables: one for animal studies and the other with human studies and discuss the results accordingly. Consider adding these as supplementary materials.

We were particularly pleased with this comment, as it shows your understanding and interest in our analysis. We have divided the table accordingly and we will attach them as supplementary materials.

  • It is mentioned in the result section that 31 out of 46 studies show significant results, these studies should be highlighted in the tables since this seems to be the results worth making conclusions from.

Again, we thank you for your helpful comments. We have now added a column to both our new Tables which refer to the statistical analysis conducted by the authors of the included studies.

  • Figure 2 needs to be in a higher resolution, it is difficult to see the text on the Y-axis. This figure seems to be of importance and should contain the proven results (significant findings).

As for the tables and the layout, we will now provide a new higher-quality image to better appreciate and understand the results. It is specified that the graph summarizes the results only of significant studies.

  • Consider structuring the discussion to reflect figure 2. Suggest that markers are grouped according to their putative timeframes (e.g 0-6 hours; 06-24 hours etc..) instead of grouped according marker type.

We thank you for your kind suggestion. While this would probably be the best way to synthesize our result, we had to keep in mind the fact that most of these markers act together with one another and, for this exact reason, most of them are commonly investigated together. We tried to add a few sentences at the end of each subparagraph to kind of summarize and explain the implications of these observations. We kindly value your opinion so let us know what you think about these modifications.

  • The reader is left without any clear conclusions regarding which markers or combination of markers that are useful when dating skin lesions, and if there are any recommendations regarding how the sampling should be performed. The conclusion could be formulated around the findings and the use of the findings, a summary figure similar to figure 2 would be constructive to show the recommended markers/combination of markers (only significant data included).

Thanks to your observation, we have realized that we were missing come more meaningful conclusions. Therefore, we have now added a few sentences that better highlight the major findings of our study.

Reviewer 2 Report

Comments and Suggestions for Authors

Well done article about a useful topic. Only some suggestions. Table 1 would be most suitable for attachment, since it is too long. t I disagree that the term lesion can be applied to a post-mortem alteration. this term should be used for ante-mortem and peri-mortem (vital) alterations. A post-mortem alteration is an artefact. Apart from that, the article is useful and the review is well done, although it is obviously too long.

Author Response

Dear Mr. Dusan Vukelic,

We would like to thank you for allowing us to submit a revised draft of our manuscript titled Wound age estimation by immunohistochemistry and immunofluorescence techniques: A systematic literature review.

We deeply appreciate the time and effort that you and the reviewers have dedicated to providing your valuable feedback on our manuscript. We are grateful to the reviewers for their insightful comments on our paper.

Following the reviewer’s suggestions, we have made some changes to our manuscript.

Here is a point-by-point response to the reviewers’ comments and concerns.

Well done article about a useful topic. Only some suggestions. Table 1 would be most suitable for attachment, since it is too long.

We thank you for your kind compliment, as well as for your observation. As you and Reviewer #1 kindly suggested, we have now listed the table(s) as supplementary material, to avoid an excessively long manuscript which is also difficult to consult.

I disagree that the term lesion can be applied to a post-mortem alteration. this term should be used for ante-mortem and peri-mortem (vital) alterations. A post-mortem alteration is an artefact.

We chose to use the term lesion also for post-mortem alterations as in our experience we have found that sometimes the time of production of the wound is not clear, i.e., it is difficult to assess whether a wound was produced before or after death. Nonetheless, the literature we reviewed also uses the term lesion for post-mortem alterations.

However, we have now made some changes according to your suggestion.

Apart from that, the article is useful and the review is well done, although it is obviously too long.

We thank you again for your observations and we hope to overcome our previous flaws with the revised manuscript.

Round 2

Reviewer 1 Report

Comments and Suggestions for Authors

Thank you for this highly improved version. However, the work still doesn’t fulfil the criteria of a systematic review and should be named a scoping review. The authors claim that each study underwent a methodological evaluation that should be presented with the set criteria´s together with the outcome for each individual paper. Stating that PRISMA standards have been employed does not automatically make it a systematic review. The criteria for being a systematic review requires transparent presentations of all the steps and evaluations. How do the authors (and readers) know that the selected search terms have located all the relevant studies in the field?

Comments on the Quality of English Language

We suggest that the journal looks over the language prior to publication.

Author Response

Reply to reviewer

Reviewer 1:

1)“Thank you for this highly improved version. However, the work still doesn’t fulfil the criteria of a systematic review and should be named a scoping review. The authors claim that each study underwent a methodological evaluation that should be presented with the set criteria´s together with the outcome for each individual paper. Stating that PRISMA standards have been employed does not automatically make it a systematic review. The criteria for being a systematic review requires transparent presentations of all the steps and evaluations. How do the authors (and readers) know that the selected search terms have located all the relevant studies in the field?”

We express our gratitude to Reviewer 2 for their observations and clarifications. In accordance with the suggestions provided by them, we have revised the title and references in the manuscript, incorporating the term 'Scoping' to 'Systematic.' The modification has been highlighted in red within the text. Indeed, we appreciate the Reviewer's insights and thank them for their prompt attention.

2)” We suggest that the journal looks over the language prior to publication.

We appreciate the reviewer for the observation. As indicated by them, a review of the English in the text has been conducted.

The Corresponding author
